# Perspectives on Citizen Engagement for the EU Post-2020 Biodiversity Strategy: An Empirical Study

**Liisa Varumo [1],\*, Rositsa Yaneva [2] , Tarmo Koppel [3], Iida-Maria Koskela [1], Mari Carmen Garcia [4], Sara Sozzo [5], Eugenio Morello [6] and Marie-Christine Dictor [7]**

1   Finnish Environment Institute, SYKE, Latokartanonkaari 11, 00790 Helsinki, Finland; iida-maria.koskela@ymparisto.fi
2   Forest Research Institute—Bulgarian Academy of Sciences, 1756 Sofia, Bulgaria; r.s.yaneva@gmail.com
3   Department of Labour Environment and Safety, Institute of Business Administration, Tallinn University of Technology, Ehitajate tee 5, 12616 Tallinn, Estonia; tarmo@koppel.ee
4   MCG Research & Innovation Sustainability Architecture/Urban Planning, 30004 Murcia, Spain; maricarmengarcia.archt@gmail.com
5   DISAFA, Department of Agricultural, Forestry and Food Sciences, Università degli Studi di Torino, 10095 Grugliasco, Italy; sara.sozzo@unito.it
6   Department of Architecture and Urban Studies, Politecnico di Milano, Milan 20133, Italy; eugenio.morello@polimi.it
7   BRGM, Strategic and Research division, 3 Avenue Claude Guillemin, BP 36009, 45060 Orléans, France; mc.dictor@brgm.fr
\*   Correspondence: liisa.varumo@ymparisto.fi

**Abstract:** The current European Union biodiversity strategy is failing to reach its targets aimed at halting biodiversity loss by 2020, and eyes are already set at the post-2020 strategy. The European Commission is encouraging the active role of citizens in achieving policy objectives in the coming years. In this paper, we explore ways citizens discuss their priorities regarding biodiversity and abilities to influence environmental problems at individual, collective and policy levels. We also examine how the citizen discussions resonate with scientific environmental priorities and how researchers see the role of citizens in policy processes and harmonising citizen and scientific knowledge. To pursue the citizen voices, an expert working group acting as knowledge brokers, facilitated a series of citizen workshops in seven European locations and a reflective researcher workshop in Belgium. Based on the results, participants identified many concrete and value-related measures to stop environmental degradation. The environmental priorities differed between citizens and scientists, but not irreconcilably; rather, they complemented one another. Both groups stressed environmentally minded attitudes in individuals and policy. Displaying diversity of perspectives was regarded as positive and adding legitimacy. Improving methods for balanced encounters among science and society is central for participation to become more than rhetoric in the EU.

**Keywords:** EU post-2020 biodiversity strategy; societal engagement; citizen participation; science-policy-society interface; public perspectives; key messages; environmental policy

---

## 1. Introduction

The current European Union (EU) biodiversity strategy, comprising six ambitious targets aiming to halt biodiversity loss and the degradation of ecosystem services, while enhancing the EU contribution to preventing global biodiversity loss by 2020, is regretfully failing to meet the majority of these targets [1]. The decline of biodiversity and its alarming effects globally have also been highlighted recently by prestigious expert groups such as the Intergovernmental Science-Policy Platform on Biodiversity

and Ecosystem Services (IPBES) [2]. Similarly, the general public is showing a rising concern for environmental issues as portrayed in the recent Eurobarometer [3] and the Special Eurobarometer 481 "Attitudes of Europeans towards biodiversity" which found that a majority of Europeans consider that we have a responsibility to protect nature for several reasons [4]. By contrast, however, only 41% of Europeans know what the word "biodiversity" means [ibid.], reflecting the different vocabularies policy and society use to discuss the environment, but not limiting the fact that citizens can still express their objectives and concerns regarding it and construct meaning for it in their societal context [5–7].

The way citizens discuss their priorities related to the environment is understandably different from the language of EU biodiversity policies, but this does not necessarily mean that the priorities and objectives are ultimately distinct. There is an apparent shared sense of urgency among science, policy and the general public for halting biodiversity degradation. The perception of many European citizens is that humanity is creating an unsustainable tension on the capability of the Earth to "absorb" the impacts derived from human activities [8]. Environmental challenges such as climate change and biodiversity loss are symptoms of a deeper problem [9]. Human demands are exceeding the absorbing and productive capacity of our planet [10], and the pressures on many of the planet's ecosystem services are close to a critical point [11]. To achieve fundamental changes in our economic and political systems and in our way of life on this planet, cooperation, communication and education without linguistic, social, ethnic or gender barriers are needed [12].

The EU has aimed to strengthen shared ownership of environmental issues and reaching sustainability and encourage collaborative action by increasing citizen involvement and participatory processes in the formulation and implementation of EU strategies [13]. The recently elected European Commission, largely responsible for the post-2020 biodiversity strategy, has noted the importance of European citizens in EU policy and legislation development and calls for more transparency and inclusiveness of the EU [14]. Including diverse types of knowledge from scientific to local and indigenous knowledge has also been recognized as important for better evidence-informed policy and achieving common environmental goals [15,16]. Several studies have been done regarding stakeholder participation in environmental management and planning from different perspectives (e.g., [17–19]) and the value of inclusiveness has also been questioned critically noting the possible disadvantages and burdens related to it [20–22]. The literature has also focused on the study of citizens' awareness of the environment and attitudes that guide their management recognizing that the study of stakeholder perception is useful for policy and practice [23]. As mentioned above, the language between science, policy and the general public is blatantly different, though ideally this should not hinder collaboration. Understanding how citizens value and perceive the environment is essential for better policy communication and participation and gaining support for decision-making [24]. Acknowledging diverse social constructs and discourses of nature as influencing the way we relate to the environment is also important to comprehend and discuss different ways of managing and approaching our environment [25,26]. In recent years the field of conservation psychology has emerged to shed further light on questions regarding what motivates individuals and communities to act more pro-environmentally and how their understanding of nature and human relations could be improved [27]. Citizen and stakeholder participation provide opportunities for social learning, which, in addition to triggering ideas and solutions benefiting from diverse perspectives and areas of expertise, may increase participants' understanding of complex environmental problems and different perspectives and interests concerning environmental management, encourage critical scrutiny of the underlying reasons of environmental degradation, and improve participants' individual and collective competencies to engage and tackle environmental challenges also in other contexts [28,29].

Participation encompasses many forms of citizen involvement, and some of them are more present in environmental sciences than others. For example, inclusion of local people and communities in collecting and observing data on natural resources and the environment in their daily lives, can be described within the concept of citizen science. Such participatory approaches of the public in scientific research may emphasize issues relevant to local communities and has the potential to fundamentally

change the relationship between science and society [30–32]. Extending this line of thinking to include citizens as legitimate knowledge and information providers is important for the legitimacy of policy [33]. It also seems evident that enabling participation alone is not enough for motivating action but rather truly understanding the diverse premises that people have for environmental issues is important [6]. Therefore, promoting open dialogue that allows these different expressions to be voiced between science, society and policy should be facilitated for improved policymaking.

European policy operates on a transnational level, whereas many of the studies above stress the locality and heterogeneity of how we relate to and experience nature. Therefore, there may be significant differences across Europe in what a country or society perceives as most pressing in terms of environmental topics [7,34]. Assuming that the general public finds it natural or simple to participate in supranational policy deliberation or management evaluation is rather optimistic [35,36]. Bridging the gap between these different scales and building relevance for the topics for different actors can occur through iterative dialogue [33] and forms of knowledge brokering [37]. Indeed, researchers and local level policymakers can have a role as "translators" of local-scale issues and citizen perceptions to construct common transnational policy and governance [38,39]. Despite these added efforts to improve communication across scales, increased participation is generally perceived positively [20,40]. Discovering ways to facilitate dialogue on local levels that is then communicated to research and policy is, however, challenging. In this paper, we aim to empirically study this challenge in the context of citizen participation for the post-2020 EU biodiversity strategy.

However, recognising or evaluating who are the people to be involved or what constitutes citizen participation or stakeholder engagement is often not straightforward, especially at the international scale of the EU, where "citizens" is a wide and heterogenous group of individuals [41,42]. It can be argued that in the context of EU policy all citizens are stakeholders, as all citizens are affected by EU policies [42]. This recognition further complicates the idea of the political legitimacy of participation as reaching all citizens is impossible. To address this issue the concept of civil society as a political community is often presented, but civil society only becomes an actor through organised civil society, i.e., civil society organisation (CSOs), which in turn brings about the dilemma of representation; to what extent can CSOs really represent the views or needs of their possibly very heterogenous stakeholders at the EU scale? [41]. Inclusion of individuals and also typically marginalised groups is another matter that requires increased attention in the development of participatory approaches in the EU so that participation is not perceived as solely a buzzword of EU institutions but converts into better inclusiveness [43]. The limitations and challenges of citizen participation at the EU scale are manifold, but for this paper we understand citizenship as denoting the right to all to be involved in civic and political affairs in the EU [44], and therefore by citizen refer to any person residing in the EU and affected by EU policy.

As recognised, solutions to global environmental problems and reaching sustainability require cooperation and knowledge sharing among diverse stakeholders, yet synthesizing their inputs and knowledge is not straightforward [45]. Knowledge synthesis refers to gathering knowledge from different sources whereas integration is the challenging task of presenting the knowledge in a concise and relevant way without fading out the core messages. Science has developed various methods to synthesize evidence and knowledge from diverse disciplines for improved management and decision-making [46], but it is apparent that there are still several challenges in harmonising and integrating knowledge from diverse disciplines and types of knowledge sources [16]. Synthesizing and upscaling non-scientific local knowledge has been researched relatively little despite the objective of many organisations and panels to integrate diverse forms of knowledge. In this paper, following the objectives of the knowledge synthesis Horizon 2020 project EKLIPSE ("Establishing a European Knowledge and Learning Mechanism to Improve the Policy- Science- Society Interface on Biodiversity and Ecosystem Services") [15] we attempt to build a flexible and light model for societal engagement for EU biodiversity issues, and even go a step forward to strengthen the science-society interface by

conducting a first attempt to harmonise societal and scientific inputs on the EU biodiversity strategy in a way that allows the plurality of perspectives to be voiced.

Biodiversity degradation and climate change as global problems need local solutions and upscaling of those solutions that are recognised as functioning and practical. Therefore, the aim of this paper is to identify how European citizens discuss environmental issues and their perceptions on how they can influence them on individual and collective levels. In addition, to shed light on the development of the post 2020 EU biodiversity strategy, we explore how these perceptions resonate with the expert opinions on the topics found central for the new strategy. Finally, we ask how researchers and citizens see the role of citizen participation in EU biodiversity policy development and how science and societal inputs can be harmonized.

The paper is structured as follows: after the introduction we present in detail the different materials collected through workshops and surveys and their analysis methods. In section three we describe the results of the different workshops and in section four we analyse the results reflecting on our research questions. Finally, we end with brief conclusions on our work and possible future research needs.

## 2. Materials and Methods

One function of the EKLIPSE project is having open "calls for requests" that may be answered by any organisation that has knowledge needs related to a biodiversity of environmental issue of European policy relevance. This request for gathering input for the post-2020 EU biodiversity strategy was put forth to EKLIPSE by ALTER-Net (alter-net.info), a network of European research institutes focused on environmental issues. Their request was that EKLIPSE would, by using their diverse methodologies for knowledge synthesis, gather input from scientists and societal actors to make a significant and policy-relevant contribution to the post-2020 strategy. For the purpose of societal and citizen input EKLIPSE launched an open "call for experts" to design and execute a process to engage citizens in different parts of Europe and eventually work towards integrating in some form the views of citizens and scientists. A prerequisite to be elected was experience on citizen participation and academic expertise. Based on this call, the societal expert working group (EWG), now authors of this paper, was selected by EKLIPSE project members. The team was coordinated by L.V. from the EKLIPSE project.

The EWG had online meetings weekly from April to June 2019 to design the process of engaging citizens. Based on previous experiences in the EKLIPSE project of trying virtual discussion events where anyone from Europe could participate to discuss an environmental topic [47] we decided this time to prefer local-scale face-to-face events that would then be synthesised by the EWG. We chose workshops as the format for the local engagement events to answer the request of synthesising societal input on post-2020 biodiversity issues. Carefully designed workshops as a research methodology serve to fulfil a double purpose: they allow the participants to express and discuss their personal views on the subject and possible learn from other participants and simultaneously contribute to the researcher's objective of obtaining reliable and valid data on the topic of the workshop [48]. The facilitator has an important role in allowing the space for each participant to voice their views without being judged and creating a comfortable and welcoming environment [49]; hence, before the workshops, the EWG discussed how to run the workshops and how to prepare for possible unforeseen issues. Knowledge about the local culture and context is also relevant for the facilitator to enable this safe space of sharing and exchanging views and knowledge [37].

Eventually the EKLIPSE expert working group organised and facilitated eight workshops in Europe: one in Bulgaria, Finland, France and Spain, and two in Italy (Milan and Turin) and Estonia (Tallinn and Tartu). The two Estonian workshops are analysed in this paper as one, since they were facilitated by the same expert, who summarised their results into one report; thus, we discuss a total of seven workshops. The workshops were organised by us as national members of the EKLIPSE EWG on societal engagement between the end of May and early June 2019 in the language of the respective country. Besides the local expert as principal facilitator, there were assistants to take notes and help in

the facilitation of the workshops. No prior knowledge of biodiversity or EU decision-making was required from the participants.

The Helsinki workshop was a pilot, and therefore only had four participants, whereas the other workshops had between 8 and 32 participants each, totalling 101 participants. The participants' ages ranged from 19 to 101, and there were 61 women and 40 men. Workshops in Helsinki, Sofia and Milan had a more homogenous audience, whereas the other workshops had a mix of participants, such as young people, lawyers, students, activists, engineers, etc. (Table 1). The workshops were open for anyone to participate and the experts used different methods to promote them and invite people, including social media, email lists, flyers, mentions in local newsletters etc. Despite it being an open workshop, due to the location and possibly other factors such as timing, promotion methods of the workshops, etc., there was a participation bias towards women and urban residents. Representation of citizens is always challenging, especially on broad issues and scales as noted above and the possible limitations of our approach are further reflected in the discussion. However, since we acknowledged these challenges prior to the workshops, the objective was never to gather a representative sample of the local or EU citizens but rather display the plurality of views and ideas within and between the different workshops. For this aim we paid special attention to the format and facilitation so that it would allow everyone to voice their opinion in written form if they were uncomfortable to do so by speaking.

**Table 1.** Overview of the citizen workshop participants.

| Workshop Location | Helsinki, Finland | Turin, Italy | Tallinn/Tartu, Estonia | Milan, Italy | Orleans, France | Sofia, Bulgaria | Murcia, Spain |
|---|---|---|---|---|---|---|---|
| Attendees (including facilitators) | 7 | 32 | 38 | 13 | 13 | 9 | 8 |
| Ages | N/A | 19–101 | 18–75 | 19–42 | 30–77 | 19–32 | 20–65 |
| Sex F/M | 85/15% | 47/53% | 62/38% | 62/38% | 80/20% | 80/20% | 65/35% |
| Prevailing profession | Retirees, nurse, researcher | mixed; lawyers, journalists, activists etc. | mixed; teachers, educators, nurses, entrepreneurs, specialists and others | students | mixed; scientists, lawyers, activists etc. | mixed; university students and early career professionals | mixed; students, civil servants, engineers, teachers, business |

The structure of each workshop was the same: after a brief introduction of the EKLIPSE project and the current EU biodiversity strategy, we presented a video by EKLIPSE (youtube.com/watch?v=BFiICakdnQw&t=2s) based on a poll of environmental concerns across Europe. Thereafter a discussion facilitated by the local experts began with sharing views on the meaning of the term "biodiversity" to appreciate the level of participants' knowledge. Afterwards, the participants were split into two to four smaller groups depending on the overall number of people to discuss i) individual actions, ii) collective actions, and iii) political actions to combat environmental degradation. In parallel to the discussion each member also filled in a survey sheet (Supplementary material Template S1) with one or two open questions related to these topics and some Likert-scale statements (Supplementary material Table S2). They were also asked to evaluate their level of expertise on biodiversity issues from 1 ('not at all') to 5 ('I'm an expert'), the average of all participants being 3, to have an overview of how knowledgeable our participants were from their own perspective. Lastly, the participants were asked to list one to three actions relevant to the post 2020 EU Biodiversity Strategy, something we call the societal key messages (KMs).

First, the results of the local citizen workshops were all reported using the same report template (Supplementary material Template S3) to ensure coherency and facilitate the synthesis analysis (Section 3.1). Besides describing the content of the workshops, the reporting also included evaluations on the general atmosphere, dynamics and knowledgeability of the participants of the workshops. The local workshop analysis was based on the surveys collected from the participants, the notes of the assistants and a discussion with the facilitation team regarding the workshop. Content analysis was

used to extract topics or themes from these materials. Qualitative thematic content analysis is well suited to the analysis of workshop materials as it aims to make inferences, improve understanding of the question studied or describe the characteristics of the content in a systematic way [50–52]. By categorising the discussion material under themes it becomes more comprehensible and allows the discovery of patterns [51]. After gathering all the citizen workshop reports, every EWG member read all the reports and then jointly discussed them to ensure that we had categorised our workshop themes in a similar and systematic manner [51]. The collection of reports was then analysed collaboratively again with content analysis to extract the similar and diverse themes that were discussed in the workshop and now presented in this paper. Based on the common themes arising from the workshops a model for biodiversity conservation from the citizen perspective was elaborated (Section 3.1.4).

Another team of EKLIPSE (see Gosselin et al. 2019 in this same special issue) worked in parallel with our EWG on gathering scientific key messages for the biodiversity strategy from researchers. As part of our work on evaluating the opportunities and challenges related to dialogue and knowledge exchange across science, policy and society and different geographical scales we organised a reflective workshop (Figure 1) to examine the societal KMs against the scientific KMs. In this workshop our EWG was to act as a type of knowledge broker or intermediary, facilitating knowledge exchange between the citizen views and the researchers. Often knowledge brokering is related to the ideas of exchanges between knowledge producers and users where the end-user is often policymakers or practitioners and the knowledge brokers may have a very formalised role [37,38,53]. In our reflective workshop and the interpretation of it afterwards we had a lighter approach to this intermediary role as the idea was to deliver the ideas of citizens to the researchers to that they could further elaborate on them and then for us to reflect the ideas of the scientific key messages back on the societal KMs. This way it was to be the EWG that produces the iterativity between the different types of knowledge-holders [37] rather than direct dialogue between the two groups. This approach was chosen to highlight the dependence of scientific knowledge production on sources of non-scientific expertise, encourage knowledge co-production and blur the dichotomy between knowledge producers and users [37,40].

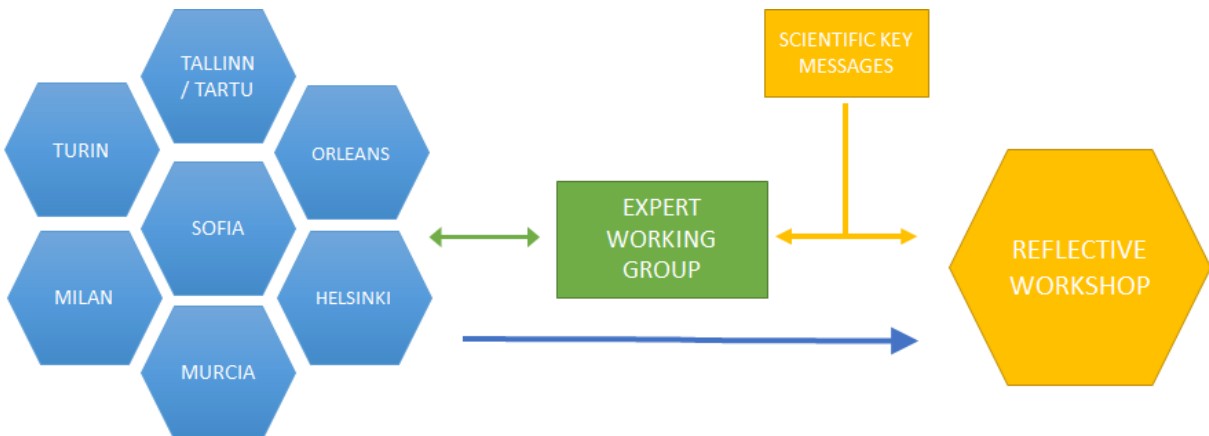

**Figure 1.** The process of data collection. Starting with citizen workshops in seven locations, the outputs (blue arrow) of which were discussed at a reflective workshop. Afterwards, the scientific key messages, also discussed in the reflective workshop, were analysed by the expert working group (yellow arrows) and interpreted against the results of the citizen workshops (green arrow).

The reflective workshop was organised as part of the ALTER-Net conference in June 2019, which had the topic "Post-2020 biodiversity targets" and had 15 participants and 6 facilitators. The participants were mainly researchers from different disciplines and practitioners involved in biodiversity policy. For the purpose of the workshop the societal KMs were presented as summaries from the three levels of actions: individual, collective and political, and the same video as in the citizen workshops was also displayed. The participants discussed, in three groups, the topics of i) how to make the strategy process

more inclusive and ii) how to harmonise scientific and societal key messages. The group discussions of the reflective workshop were recorded and analysed by the EWG using content analysis (see above).

In addition to discussing the societal KMs with researchers we also cross-evaluated the content of the citizen workshops in how they resonate with the scientific KMs. This was done by comparing the local workshop reports and discussions with the list of the scientific KMs and evaluating whether the topics or themes of the scientific KMs had been mentioned in the citizen workshops on a scale of not mentioned-low-medium-high. High mention meant that the topic was discussed various times in the workshop and was also prevalent in the written surveys, medium that the topic was discussed or written about moderately and low that it was mentioned as a kind of side-note. The list of scientific KMs is from June 2019 and has evolved since.

## 3. Results

In this section, we report the results from the citizen workshops and the three levels of action (individual, collective, and political) discussed in them and summarise the results in a model displaying the connections between science, policy, society and biodiversity loss and conservation from the citizen perspective. We also present how the discussions of the workshops reflect the topics visible in the societal KMs. Chapter 3.2 focuses on the outcomes of the reflective workshop in Ghent.

### 3.1. Citizen Workshops

#### 3.1.1. Individual Actions

Individual actions inspired the most suggestions of the three levels of activity. The individuals' measures for protecting biodiversity in the workshops included concrete everyday actions and personal choices that people could do as well as many value-related general views or ways of thinking about and relating to biodiversity (Table 2).

Consumption was the overall theme under which most of the individual measures can be allocated. Consciousness about personal consumer choices and material issues were discussed to some extent in all the workshops. In connection to this, the topic of food was mentioned in almost all the citizen workshops. In particular, the idea of favouring locally and organically produced food was highlighted in six out of seven workshops. Increasing the amount of vegetarian food was especially talked about in Helsinki and acknowledging issues of food waste and the whole life cycle of food production was discussed in Milan, Murcia, Turin and Sofia. In addition to food waste, six workshops highlighted waste, environmental pollution and recycling in general. In this context, reducing and reusing materials, especially plastic, was brought up in Murcia, Milan, Sofia, and participating in the activities of waste removal in natural areas were mentioned in Orleans and Sofia. The topic of pollution was also related to transport, particularly in Sofia, Murcia and Milan. Promoting public transport and carpooling or sharing and favouring bicycling were individual pro-environmental measures mentioned in six of the workshops, not directly linked to consumption.

Topics that were less prevalent across workshops, but which were prominently mentioned in one or two locations, varied. Choosing renewable energy, gardening and using organic medications, avoiding pesticide use and chemicals in industry were among these miscellaneous themes. Some of these concrete actions were not only related to improving the condition of biodiversity but also evidently to human wellbeing, illustrating the diverse relations between wellbeing and the natural environment. This was also apparent in the initial discussions in the workshops on the meaning of biodiversity, where many explained it as the foundation of life on earth and important for humankind and future generations.

**Table 2.** Individuals actions for sustainability and biodiversity conservation. Based on the votes and discussions in the workshops (1 = most important, 5 = less important).

| Ranking | Sofia | Orleans | Turin | Milan | Murcia | Tallinn/Tartu | Helsinki |
|---|---|---|---|---|---|---|---|
| 1 | Waste sorting, including food left-overs (ask for take-aways at restaurants) | Responsible consumption (organic, local, less ethnic, repairs things, bulk food) | Food: local healthy food, less social inequality and healthy food for everyone, fight food waste | Food intake and alimentation, increase diversity and choices of foods and dietary elements | Reuse, reduce and recycle more (i.e., reduce use of cars, walk more) | Mental awareness and material needs developed in a balanced way | Recycling |
| 2 | Walk, bike and use less car transport (prefer carpooling) | Knowledge of local actors on biodiversity | Raise your children to be environmentally aware | Waste: recycle and reuse, less mono-use plastics and reduced packaging | Responsible consumption (consume only what we need and buy local products) | Raise the biodiversity issue to the global level, but recognise individuals' position in the global system | Use of public transport |
| 3 | | Gardening in a sustainable way (biodiversity spaces with insect hotels, beehives) | Recycling | Mobility: increase public transportation, bikes, carpooling and sharing, reduce emissions | Awareness about the impacts of pesticides for bees, plastic for the environment and industrial chemicals on species and human health | Shaping environmental awareness (at the heart of sustainable development and ecological lifestyles), personalities and own internal beliefs | Preferring domestic foods |
| 4 | | Waste (reduce packaging, clothing, electronics, furniture), pick-up waste, recycle | Prefer domestic foods | Defragmentation of land-use, work choices, resource efficiency, communication and awareness | | Development of individuals spiritual qualities accompanied by ethical dilemmas | Raising your children to be environmentally aware |
| 5 | | Involvement in local politics, nature cleaning operations, participatory science programs, biodiversity inventory, environmental organisations | Prefer online meeting instead of travelling | | | Be in accordance with inner values and attitudes | Preferring online meetings instead of travelling |

The value-related views on personal actions concerned the idea that people have a responsibility to be educated about biodiversity issues and share this knowledge as well as recognise human impacts on nature. The idea of environmental education from early childhood on was mentioned in six of the workshops and, relatedly, raising both personal and public awareness about the environment was acknowledged as important. The discussion in the Tallinn/Tartu workshop became highly centred around the ideas of personal spiritual reflections on humans and nature and understanding the linkages from local actions to global consequences for the environment. In Helsinki, Murcia, Orleans and Turin, political action in the form of voting or encouraging and investing in dialogue with local policymakers was also seen as a type of personal responsibility. In Orleans, participation in citizen science programs was said to be a personal way of participating in helping the environment, and in the Finnish workshop, one participant also mentioned his choice of profession as an environmental researcher to be a personal action to hinder biodiversity loss.

### 3.1.2. Collective Actions

There was more diversity in the discussions and responses regarding the collective actions to halt biodiversity degradation. Many of the collective measures were either directly or indirectly linked with the individual measures or expressions on the need to promote the participation of individuals to collectively act in more pro-environmental ways and have more environmentally minded attitudes (Table 3).

Different types of voluntary activities were mentioned in majority of the workshops. Forms of voluntary work included city or neighbourhood organized groups to clean up the immediate environment, neighbourhood projects to revegetate urban spaces and plant urban gardens and teach gardening to promote local food production. Additionally, volunteering in some type of association or organization, such as the scouts, that promotes environment-related activities, and respect for the environment was regarded as a collective measure. In Finland, many of these collective measures were related to activities that you carry out within your housing cooperative, whereas in Orleans, spreading them to even a regional level to ensure a broad involvement of people was considered to be important.

Another theme that was commonly mentioned in most workshops was some form of raising awareness of and mainstreaming biodiversity. This included, for example, gathering people to organize public campaigns or seminars to spread information and engage local communities in environmental and ecological topics. Also, improving the communication between decision-makers and citizens and spreading awareness on how to participate in political action and influence policy and legislation was mentioned explicitly. Different types of bottom-up and from local to global approaches were discussed in some of the workshops.

In three workshops, the elements of circular economy and recycling were mentioned, concretely the sharing of products such as books, tools, toys, etc., exchanging seeds and carpooling were discussed as collective actions. Teaching people to reuse old items was also mentioned in one workshop.

**Table 3.** Collective actions for sustainability and biodiversity conservation. Based on the votes and discussions in the workshops (1 = most important, 5 = less important).

| Ranking | Sofia | Orleans | Turin | Milan | Murcia | Tallinn/Tartu | Helsinki |
|---|---|---|---|---|---|---|---|
| 1 | Proactive behaviour at home, work and school | Sharing (tools, books, toys) and spaces like gardens, exchange goods like seeds, cars etc. | Local production collectively, goods for all and no food-waste | Local production and circular economy | Stand up against consumerism and favour e.g., sharing with neighbours | Organise seminars, communicate necessary changes and foster interest in the environment | Active participation in housing community for recycling, cleaning the environment, etc. |
| 2 | Volunteering in public campaigns | Develop neighbourhood projects (land-shared gardens, collective urban cleaning and recycling, DIY workshops) | Change environmental policies with help of scientists informing policy and inclusion of citizens | City scale: change policies and re-plan cities while overcoming political resistance often faced when breaking silos, the role of EU to support new policies | Spread awareness of ecological issues | Environmental awareness as a guideline for life quality, detach from consumerism | Joining organisation or events that are pro-environmental |
| 3 | | Citizens' lobby: legislative proposal network of influence, communicate actions with transparency and expert facilitators, condemn greenwash practices | Interconnected communities of interest to help monitor and disseminate information on environmental challenges caused by biodiversity loss and climate change | Communities of interest to monitor and diffuse information on the environment and biodiversity loss and citizen science | | Society to support new ways of thinking. Human development important for environmental awareness, belief in old values (justice, peace, respect for life, aware behaviour) | |
| 4 | | Involvement in collective activities like revegetation of urban areas, awareness raising activities | Ask media to communicate better to citizens on environmental issues | Ensure dissemination of correct information try and uphold the trend of environmental issues which now is fashionable but may not be so in the future | | Campaigns to promote better values in a creative way in different situations, study programs etc. | |
| 5 | | Provide scientific understanding on political decisions | | Foster inclusion | | | |

### 3.1.3. Citizen Expectations towards Policy

The political actions were discussed mainly as expectations or suggestions that the participants had towards different scale policymakers from local to EU. There were some differences between the cities regarding how comfortable the participants felt talking about politics, the participants of Sofia being somewhat reluctant to discuss these issues (Table 4).

The point of using different types of knowledge, mainly expert but also local citizen knowledge and information, as a basis for policymaking was stressed in all of the workshops. In this context, the idea of providing training for decision-makers to improve their understanding of, e.g., different scales of policymaking and long- and short-term objectives were mentioned together with consolidating structures that enable policymakers to have relevant expertise at hand when making decisions. Increasing the accountability of policymakers was addressed in half of the workshops. Asking policymakers to report on progress to show that promises and commitments are being achieved was a concrete suggestion to improve accountability.

Training or awareness-raising was also mentioned at the policy level for citizens as something that should be supported by policy in majority of the workshops. The participants of Tartu/Tallinn workshops expressed that this type of awareness-raising should be based on national values to design appropriate environmental policies. Improving transparency and the clear and understandable communication of policy-issues, especially from the EU scale, was called for in four of the workshops. In Turin, the idea was expressed that better-informed citizens are more environmentally friendly citizens.

Another popular topic in the workshops was the idea of hard measures to prevent environmental degradation. These included removing harmful subsidies and incentives and developing clear and strict regulation and legislation that would penalize and enforce sanctions on actions that have negative impacts on biodiversity.

In the Murcia workshop, people also saw voting for politicians or parties with pro-environmental agendas as a political action and in Tallinn/Tartu the participants mentioned that the mindset of the government needs to be shifted towards a more sustainable direction. They also stressed that many people were ignorant or uncaring with regard to the environment, and it is exactly these people that the government must find ways to push towards more sustainable lifestyles through hard and soft measures.

### 3.1.4. Biodiversity Conservation Model

Derived from the citizen workshops, Figure 2 presents a model of how science, policy and society are interlinked and how they both enable biodiversity loss and explore methods to improve biodiversity conservation. The outer elements depict societal values and scientific research, where societal values are based on both individual and community values on how nature is regarded. Both societal values and scientific research would ideally induce corresponding policy settings, which is a basis for environmental policy. However, this is often perceived to not be the case (implied by the dashed line). When environmental policy lacks an effective means of sustaining biodiversity, combined with harmful everyday actions of social stakeholders, the impact on biodiversity is evident. As suggested in the workshops, environmental policy could prevent some of these everyday actions with legislation and removing harmful incentives. The general mean to conserve biodiversity was identified to be environmental awareness—when social actors become aware of their actions and the impacts of their everyday habits, supported by corresponding guiding principles, derived from best practices presented in scientific research. Thus, a combined and synchronized effort is required to successfully stop biodiversity loss.

**Table 4.** Policy expectations for sustainability and biodiversity conservation. Based on the votes and discussions in the workshops (1 = most important, 5 = less important).

| Ranking | Sofia | Orleans | Turin | Milan | Murcia | Tallinn/Tartu | Helsinki |
|---|---|---|---|---|---|---|---|
| 1 | Increase public awareness | Training for decision-makers, elected officials by an organisation such as IPBES | Raise environmental awareness via strong cross-cutting action plan with different groups to overcome ignorance | Achieve a more persuasive message on the environment: everyone should talk about it | Bring external environmental expertise to government | In-service training as a form of lifelong learning for a wide range of stakeholders | Diverse possibilities of policy participation for diverse groups, ages, etc. |
| 2 | Volunteering in public campaigns | Citizen jury participation in impact studies, e- petitions, referendums pre-projects, participatory democracy training, public hearings w/ policymakers | Incentives for better farming techniques and local production | Create strong networks and promote inclusion of all society | Importance of political communication for environmental awareness | Change the government mindset to more pro-environmental and combating climate change | Becoming more informed on what happens in the EU via e.g., more understandable news on current affairs in local media |
| 3 | | Accountability with financial penalties and incentives for biodiversity protection in upstream production | The citizen must feel personally involved in political action, but to feel involved he/she must have understanding | Act on perceptions of relevance of the challenge of environmental preservation | Vote for parties that embrace environmental values and have sustainability on their agenda | Political support for environmental awareness, with focus on national values. Design a value-based society and aware nation | Sanctions and accountability for policymakers and industry |
| 4 | | Invest in education of youth (schools and other activities) | Correct dissemination of information | Pressure decisionmakers to fulfil promises, monitoring progress of promises and agreements | Need for more regulation in sectors dealing with the environment and natural resources | Change environmental culture and ethics. Awareness as a tool for conservation of nature and wildlife | Voting |
| 5 | | Reform media | | | | | |

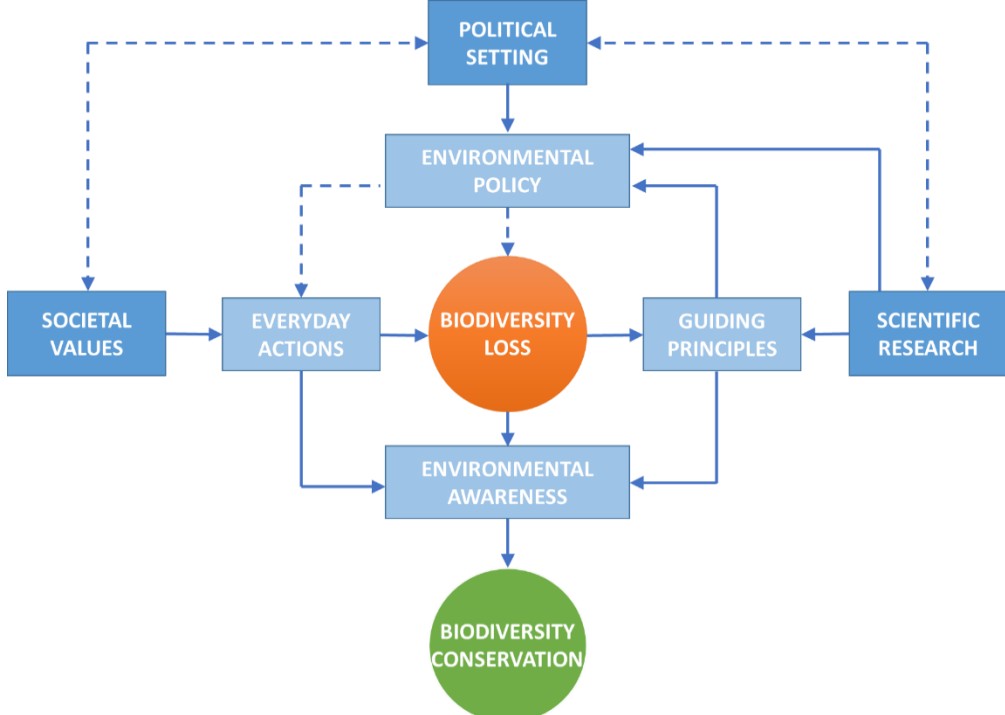

**Figure 2.** The model for improved biodiversity conservation based on the citizen workshops.

### 3.1.5. Scientific Key Messages and the Citizen Workshops

The reflection by our expert working group on the topics discussed in the citizen workshops and their relation to the scientific key messages gathered from researchers illustrated that there were some shared themes among the two groups, citizens and scientists, but also many messages that were not present in the citizen workshops (Spreadsheet S4).

The discussion in six of the workshops revolved highly around diverse ideas of changes in society's mindset for more pro-environmental policy and actions (KM05) (Table 5) and in line with this, recognising the dependence of society on biodiversity and natural resources (KM04) was also acknowledged in five of the workshops. Using diverse forms of knowledge (KM08) and increasing participation (KM11) were also central in five workshops. Monitoring and evaluation to gain better information and mainstream biodiversity across sectors (KM02) and the ideas of decoupling economic growth from biodiversity degradation (KM10) by, e.g., eliminating harmful subsidies were also discussed in about half of the citizen workshops.

Comprehensive policy mixes to halt biodiversity loss (KM15) were not mentioned in any of the citizen workshops. Similarly, freshwater biodiversity (KM16) was only mentioned in one workshop, and KM12 on incorporating policies and management on longer temporal and larger geographical scales was barely mentioned in any of the workshops. Interestingly, climate change (KM01) as interlinked with biodiversity loss was also absent or rarely mentioned in many workshops.

**Table 5.** The scientific key messages and the mentions of their themes in the citizen workshops on the scale high–medium–low–not mentioned (N/A). Acronyms: biodiversity (BD), ecosystem services (ES), nature-based solutions (NBS).

| Scientific Key Message | Helsinki | Turin | Tallinn/Tartu | Milan | Orleans | Sofia | Murcia |
|---|---|---|---|---|---|---|---|
| KM01 \| Climate change | Low | N/A | Medium | Medium | Medium | Low | High |
| KM02 \| Monitoring and evaluation | High | Medium | Medium | High | High | Medium | Medium |
| KM03 \| Core drivers of BD loss and integration across sectors | N/A | Medium | N/A | N/A | High | Low | High |
| KM04 \| BD and ES a condition for human activities and quality of life | High | High | High | High | High | Medium | N/A |
| KM05 \| Mind-set changes for BD and ES | High | High | High | High | High | Medium | High |
| KM06 \| Restoration of ecological functions | N/A | N/A | High | High | Medium | Medium | Medium |
| KM07 \| European and global policies | N/A | N/A | Low | Low | N/A | Medium | Medium |
| KM08 \| Research and knowledge-informed decision-making and implementation | High | High | High | Medium | High | High | N/A |
| KM09 \| Importance of multi- inter- and trans-disciplinary research | N/A | High | High | N/A | Medium | N/A | N/A |
| KM10 \| Decoupling economic growth from environmental degradation | N/A | High | Medium | High | High | Medium | N/A |
| KM11 \| Increasing participation and stakeholder involvement in management | Low | N/A | High | High | High | High | High |
| KM12 \| Incorporate regional/transnational processes and long-term temporal scales | N/A | N/A | Low | N/A | Low | N/A | High |
| KM13 \| NBS and conservation for sustainable development | N/A | Medium | High | Medium | Medium | N/A | Medium |
| KM14 \| Inter- generational sustainable transformations | Low | High | Low | Low | Medium | Low | Medium |
| KM15 \| Comprehensive BD policy mixes | N/A | N/A | N/A | N/A | N/A | N/A | N/A |
| KM16 \| Freshwater BD | N/A | N/A | Low | N/A | N/A | N/A | N/A |

*3.2. Reflective Workshop*

3.2.1. Reflections on Combining Research and Citizens' Perceptions

The discussion in the reflective workshop did not only concentrate on how to specifically merge the scientific and societal key messages but rather on a broader level focusing on the participants' personal experiences about interactions between science, policy and citizens. The participants recognised many benefits of citizen participation in research and policy processes, but also acknowledged that participation often remains at a rather superficial level with the premises of participation set by researchers and policy-makers and not truly happening on the conditions of citizens.

Most participants saw value in keeping the different types of messages or recommendations separate. They recognised that there were essentially no significant discrepancies between the messages, but that they rather complemented each other, and that merging them might risk losing important dimensions of the messages. No final conclusion was achieved on how they should be combined, but it was suggested that in the final output they could either be completely different documents or that the links where the messages complement each other should be made somehow visible. Making it visible that both researchers and citizens had been included in the process was mentioned as possibly adding legitimacy to the messages in the eyes of policy-makers.

It was noted that the scientific recommendations for policy are often written in overly complex, long and difficult language, whereas the societal messages were seen as more concrete and actionable, mainstreaming the future biodiversity strategy. The complexity and extent of the scientific messages was discussed as a possible drawback for both communicating them to the public but also policy-makers.

Based on the messages the participants observed that the ways research and the general public express and comprehend their concerns or ideas of biodiversity issues differs not only in language but also the scope of what is discussed. Researchers tend to have a more focused view on biodiversity as strictly something to do with species, habitats and other elements of nature, whereas the societal messages illustrate that the public is more comfortable thinking of the environment as a rather broad concept that is visible in everyday life and actions. This was given as one possible reason for why the societal messages ranged from pollution to climate change and recycling while the scientific messages seemed to centre around species monitoring, habitat protection, ecosystem services policy, etc.

3.2.2. Citizens' Role in Environmental Policymaking

In the second part of the discussions, researchers explained policymaking and science as processes which both can and should engage citizens. Several potential approaches for citizens' engagement in biodiversity research and policy processes were identified, ranging from passive knowledge assimilation and everyday behaviour to becoming active change agents urging societal transformations.

When a more passive role was assumed, citizens were mainly seen as objects of scientific information, whose knowledge of biodiversity, connectedness to nature, and motivation to act in a pro-environmental manner should be increased through, for example, educational programmes and clear communication. The uncertainty of scientific information, the various sources of information that citizens use in their daily lives, and the different levels of interest towards nature and biodiversity were seen as major challenges for effective communication and for encouraging citizens to take action for biodiversity.

Most of the experts acknowledged that for enhancing the success of biodiversity policies, it is important to involve citizens' views during policy development. Science–policy–society dialogue was regarded as a more motivating approach to ensuring that citizens understand the value of biodiversity and want to protect it than simply disseminating scientific information to citizens and 'telling them what to do'. Different methods for involving citizens' perspectives were identified, including polls capturing the views of the larger public, as well as deliberative sessions with focus groups. Also, citizen science was considered as a promising method for increasing personal interest towards biodiversity and for allowing citizens to participate in policy monitoring.

Nevertheless, politicians' responsibility in solving environmental problems was emphasised in the discussions. While engaging citizens in policy development was regarded as important and pressure from the public and different bottom-up initiatives necessary for bringing about change, political decision-making was considered as a crucial means for encouraging systemic change. Assuming the responsibility of tackling major environmental problems to citizens and their individual actions was seen as unfair, especially when it involves an economic or social burden, e.g., a necessity to pay for green behaviour. Therefore, bold decision-making was hoped for from policy-makers. Non-negotiable situations were distinguished from more unclear and complex ones, and public participation was given higher relevance in the latter case.

## 4. Discussion

The results of the citizen and reflective workshops give insights into both (i) the substance of what types of environmental concerns both groups consider as central for the post-2020 strategy and possible roles different actors or groups have in addressing them and (ii) how participation in policy processes is perceived and what challenges it contains. These two themes are discussed here to respond to our research questions.

### 4.1. Reflections on Pro-Environmental Behaviour and Differences between Key Messages

The variety of measures to halt biodiversity degradation suggested during the citizen workshops tended to build on the idea that raising awareness on biodiversity and human impacts could translate into actions in consumption, food production, recycling and environmental quality valorisation in general. This "awareness produces action" type of rationale has also been proposed as one of the reasons for encouraging participation in environmental policy development [54]. It has also been suggested that deliberative processes, such as our workshops, can be seen to foster critical self-awareness and thus lead to better environmental results [55]. However, the extent to which awareness and values truly lead to action remains contested and somewhat unclear both in theory and practice [56–58], and exploring this topic would require further data from our participants. In general, however, the ways that knowledge exchange and awareness raising translate into action is different between individuals, reflecting, e.g., cognitive barriers to environmental action and personal defence mechanisms to cope with complex and distressing information [59], as well as at collective or more systemic levels, where the interdependence between actors disables individuals to change practices without the collective [60,61]. This line of thinking was also apparent in the workshops, considering that most of the individual actions related to concrete daily measures, whereas the collective and policy measures called for more long-term changes in collective mindsets to enable pro-environmental policies and practices.

The discussions of citizens and researchers on emerging environmental issues reflect a common effort to collect more than pure data or scientific knowledge to guide policy and to gather meaning, public perceptions, and common understandings on what motivates society (individuals as well as communities) to act pro-environmentally. These motivations and values may help us understand how to better enable society to fulfil the potential of social learning that may produce solutions and competencies to strive for sustainability [28]. It was positive to see the amount of ideas produced in the workshop as this gives hope that many have not become so overwhelmed with the scope of environmental issues that it would paralyze them from seeing solutions to these issues [62].

The comparison between the societal KMs and scientific KMs confirmed the rather different approaches of the two groups to discussing biodiversity and the environment. This distinctiveness is also apparent in observing the targets mentioned in the current biodiversity strategy [63], which include topics such as invasive alien species, references to conservation directives and policies, etc., and comparing them with Eurobarometer [4] results, where climate change, air pollution and waste are seen as the top three environmental problems. These differences may be partially explained by the level of governance, where individuals may be more aware of personal or local policy actions to certain problems such as waste or air pollution that is tangibly experienced [64], whereas international policy

needs to operate on a broader scope addressing whole sectors (forestry, fisheries, agriculture in the BD strategy) that are relevant for the entire EU. Additionally, mass media is often central to shaping what citizens see as urgent and pressing environmental issues [65,66]. Considering the possible influence of the media, it was, however, interesting that the topic of climate change was rather absent in many citizen workshops, mainly discussed in Murcia where they have perhaps most been impacted by it in terms of very high temperatures in previous years. By comparing the two sets of KMs, we aimed to illustrate that despite differences in vocabulary, the actual problems may not be too different, as there are often interlinkages between them. However, illustrating to the general public how EU policy responds to the explicit concerns that they express is probably something that would enhance acceptance of it and make participation seem more meaningful to the general public [37].

Indeed, even though policy might not currently seem to directly reflect or address in a satisfactory manner some of the key issues expressed by the citizens and commonly discussed in the media [66], research has acknowledged many of them and also the role of citizens in solving these issues. Some studies also show that the gap between awareness and action or acceptance of courses of action is narrowing in Europe. For example, in local food production in the U.K., a rise of groups of "concerned consumers" favouring local products appears to be happening [67]. Similarly, in Toulouse, France a growing interest of local people towards influencing policies related to urban food governance has been identified [68]. For waste, including food waste, pro-recycling attitudes of citizen have been reported [69] and local governments have also invested in many places in studying public behaviour to find ways to motivate, softly or by enforcing legislation, citizens to contribute to achieving waste management objectives [69–71]. The sharing economy and several related concepts such as circular or community economies have also gained popularity among citizens advocating sustainable consumption [72,73]. Acceptance of eradicating harmful subsidies or imposing taxes or costs on environmentally harmful industries has also gained public support, though generally when costs are allocated to the polluter and not the individual [74,75]. These are just some examples illustrating and supporting the possible slight trends towards pro-environmental behaviour regarding some of the most popular topics from our citizen workshops. However, these still seem to remain alternative or marginalised efforts with respect to the current state of the environment in Europe [76], with many studies arguing that motivations are not always environmental [59,73,77,78], and the failure to reach the biodiversity targets [1] obviously shows that we are far from the level of awareness of and action with respect to sustainability, both at citizen and systemic levels. It is also beyond the scope of this paper to debate whether the concerns expressed by citizens are as urgent or impactful from the scientific perspective as the citizens perceive them to be, but recognising the similarities between the different citizen workshops and the acknowledgment in the reflective workshop of the worth of these expressions combined with the amount of research on these specific topics does underline the value and validity of the citizen concerns.

It is not solely the validity of the substance of the citizen concerns that makes their acknowledgment important for policy, but also the human and societal values inherent in the expressions reflecting what society identifies as important and what they see as threatened. Naturally, there may be differences in these values within and between societies. Policy changes and management can act as a solution to address these concerns and also develop ways to harmonise interests and guide the behaviour of citizens to respond to the needs of societal interests [79].

### 4.2. The Science–Society Interface and Participation in Environmental Policy

Based on our empirical results from the citizen and reflective workshops and the model for halting biodiversity loss (Figure 2), a trend towards building open and collaborative spaces for science, society and policy is coming forward. As remarked in the introduction, this trend is not completely novel, although the extent to which these types of open spaces manage to produce meaningful outputs of multiple perspectives is still questionable [47,80,81]. Also, designing participatory processes that allow for the integration of different types of knowledge has been found challenging [7]. As mentioned in the

reflective workshop, there is value in retaining diverse perspectives. Embracing and communicating the diversity of views and practices and cultural diversity can ideally lead to richer biodiversity [24,82]. A clearer interpretation of what integration at the science–society interface means in environmental policy and what its expected implications are is needed.

The efforts to improve environmental protection and halt biodiversity loss by implementing interactive processes entails some limitations and certain level of uncertainty in the outcomes projected. As noted, citizen perceptions and judgments are often based on a range of heuristics that can introduce bias into the outcome. On the other hand, experts' opinion is typically influenced by moral and professional ethics that can lead to either "under confidence" or "over confidence" [83]. Therefore, as implied in the model, ideally there would be more collaboration between actors for biodiversity conservation (Figure 2), and a reference to citizen science will place emphasis to a community-driven research, based on feasible data, expertise, and soliciting public input on issues raised not by scientists but by communities [31].

There was little conflict within and between the citizen and the reflective workshops regarding the issues and their relevance and the KMs can be seen to complement each other. Both workshops hosted participants that were probably more aware of and positive towards environmental issues than the general public in Europe. Reed et al. (2014) discuss representation as one of the five principles to be considered in knowledge exchange, referring to the idea that identifying and embedding certain stakeholders into research and considering the ethics of stakeholder engagement is important [37]. Reflecting on this principle, it can be fairly argued that our workshops did not capture citizens from diverse domains some of which may be more affected by the biodiversity policy, including landowners, businesses, rural habitants etc. and who may also hold relevant knowledge to discover and provide acceptable solutions to the environmental issues. The citizen workshops also mainly contained urban residents, as they were held in cities, possibly skewing the views on some of the topics; for example, regarding food production and waste management there can be significant differences in access or preference to products [67] or ability to influence waste management [69]. There was also a bias for women in our workshops, which could impact the actions and concerns expressed as gender tends to influence how one sees causalities between environmental problems [58]. Geographical and cultural differences, levels of education and many other socio-economic factors have also been found often to influence opinion and perception of citizens [59,77]. Any given participatory exercise aimed to prioritise topics, such as our workshops, is always a representation of the inputs of its participants [84], and therefore should not be undermined if disclosed as such. The non-polarisation of opinions can be partially explained by noting that voluntary discussion events around limited topics tend to gather rather like-minded people [85]. By observing, e.g., popular media and other sources, it is clear that environmental issues and the solutions to them are often contested and can be denoted as wicked problems [86]. Nonetheless, knowledge exchange is facilitated by low issue polarisation, as this allows participants to have a more rational and focused dialogue [60]; hence, for the purpose of the workshops, the absence of conflict was beneficial.

In the future, more focus must be placed on methods to involve and inform those who are (willingly) ignorant to environmental topics. Participation in knowledge exchanges always has a cost for the participants, hence the value of the engagement and what participants benefit from it should outweigh this cost [37]. Similarly, pro-environmental behaviour also tends to have a cost, not necessarily economic, for people if it entails a shift from current practices to something that requires for example learning new habits or practices [59]. The lower the costs, the more likely people are to adopt new practices and submit themselves to raising their awareness. Our reasons for seeking citizen participation for the post-2020 strategy was normative in the sense that we believe that participation legitimises policymaking [42], although, as mentioned above, such discussion can also have an instrumental impact on the participants and as we communicate the results to policymakers they can hopefully have an instrumental impact on policies too. Discovering engagement approaches that have a low cost for participation but are able to have both normative and especially instrumental

outcomes is very demanding but based on the citizen workshop results it is something that is highly needed to align the interests and actions of all towards a more sustainable society.

In the design of our citizen process, we wanted to create spaces where people would feel comfortable discussing the topics on their own terms, not dominated by, e.g., the framings of science [87]. Through efforts to speak in the language of the general public, we got meaningful responses and the themes did not seem too complex to discuss. Afterwards the researchers received the opportunity to reflect on the synthesised messages. This approach, where the EWG acted as knowledge brokers and facilitators of knowledge exchange [37,38], ensured that the researchers would focus on listening to the citizen perspective and not overpower it with their professional views on what is relevant or important. Producing a true deliberative dialogue on an international scale between science, policy and society can be challenging [47], and hence approaches that allow locals to freely discuss their views and where a third party knowledge broker synthesizes and communicates these views can be valuable and respond to policies needs to seek diverse responses that could encompass a variety of issues.

Our approach was also light, systematic and cost-effective to organise. Naturally some of these characteristics were due to the reasonably manageable number of participants. It seems difficult to find mid-range approaches to international engagement as methods often tend to be either fairly local and small-scale or very resource consuming large scale efforts such as the WWViews on biodiversity, which in 2012 hosted simultaneous 100-person deliberation events in several locations around the world and then synthesised the results of the individual events to present input for the Aichi targets [88]. The EKLIPSE project has evaluated diverse methods to synthesize knowledge and recognised some methods as more open to engagement [89], but many of them are more appropriate to expert consultation rather than citizen deliberation or participation and few are based on multi-phased consultations. Although the chosen method of having separate workshops for citizens and researchers did not allow for a deliberative dialogue between science and society in the sense of being reciprocal, open and inclusive of both parties simultaneously, by designing multi-phased approaches and adding an element of iterativity, e.g., continued dialogue either via knowledge brokers or directly between science and society, we can come closer to reciprocal dialogue [33] that can help to harmonise views without losing their diversity. This feature is still lacking in many EU participation process, such as the Eurobarometers, where opinions are collected regularly but not further discussed. There is also a lack of transparency on how the results of citizen consultations are used and whether they inform policy. Our results will be reported together with those of the scientific key messages and handed over to the European Commission as agreed with the participants of the workshops.

## 5. Conclusions

In the current global situation, many citizens do have a wide set of ideas on how to act more sustainably and encourage pro-environmental behaviour. The citizens presented both concrete practices and many value-based suggestions, which underscores the idea of values guiding our everyday actions and that these societal values should also guide the policies developed in democratic countries. However, more research is required to identify how and if this awareness really transforms into action and what diverse measures science and policy can produce to encourage this transformation in the heterogenous publics of Europe. Additionally, the expressions between citizens and science on the most pressing environmental issues differed, but the objective of halting biodiversity loss is similar, and the means expressed by the two groups support rather than contradict one another. The plurality of perspectives and illustrating plurality strengthens legitimacy and transparency of policies. Finding ways to show the interlinkages between the different types of concerns and expressions can thus help to produce more acceptable and sustainable policies in the EU.

Facilitating participation is not a panacea for reaching sustainability as many citizens remain voluntarily ignorant or excluded from participating in pro-environmental actions and reaching these people must be addressed to progress towards a more sustainable society. Ideally, as expressed in the model from the citizen workshops and the reflective workshop, science, policy and society

would exchange knowledge and views on policy questions for improved biodiversity conservation. Acknowledging the value of different opinions and ideas raised by science and society has been an objective in many policy processes on the international level and the post-2020 biodiversity strategy has a unique opportunity to improve on these participatory methods by also learning from the challenges other initiatives have encountered upon integrating diverse knowledge holders. Different engagement approaches appealing to different audiences and ways of gathering and synthesising input from different actors, including those often omitted or marginalised, need to be developed. A step forward is that instead of forcing the integration of different perspectives and seeking unanimity, we argue that synthesising views from local levels in systematic ways using multi-phased methods also allowing the reflection between different perspectives, somewhat similar to what is done in systematic reviews in science, we can form more comprehensive outputs to feed into policymaking. Being transparent about the uses of the diverse consultations and participatory process, both in the EU and by researchers and other organisations, is key to showing results to participants and motivating future participation.

**Supplementary Materials:** The following are available online at http://www.mdpi.com/2071-1050/12/4/1532/s1, Template S1: Workshop survey template, Table S2: Results of Likert scale statements, Template S3: Workshop reporting template, Spreadsheet S4: Scientific key messages in June 2019 before Ghent conference.

**Author Contributions:** Conceptualization, L.V.; R.Y.; T.K.; I.K.; M.C.G.; S.S.; E.M.; M.C.D., methodology, L.V.; R.Y.; T.K.; I.K.; M.C.G.; S.S.; E.M.; M.C.D.; formal analysis, L.V.; R.Y.; T.K.; I.K.; investigation, L.V.; R.Y.; T.K.; I.K.; M.C.G.; S.S.; E.M.; M.C.D.; data curation, L.V.; writing—original draft preparation, L.V., R.Y., T.K., I.K., M.C.G., S.S., E.M., M.C.D.; writing—review and editing; L.V.; R.Y.; T.K.; I.K.; S.S.; visualization L.V., T.K., M.C.D.; supervision, M.C.D.; project administration, L.V., I.K.; All authors have read and agreed to the published version of the manuscript.

**Funding:** This work was supported by the EKLIPSE project funded from the European Union's Horizon 2020 programme (Grant agreement no. 690474).

**Conflicts of Interest:** The authors declare no conflict of interest.

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
