# Peer review of "Perspectives on Citizen Engagement for the EU Post-2020 Biodiversity Strategy: An Empirical Study"

_sustainability, doi:10.3390/su12041532_

Round 1

Reviewer 1 Report

the research is interesting and is well presented. The main objective appears to be support for public policies for the conservation of biodiversity through the involvement of experts and citizens. However, if this is the objective, the "sampling" of citizens seems to be significantly unbalanced towards women and urban elites. As the social history at European and global level is showing, it is necessary that the social groups of the rural areas and the weaker social groups are also taken into account in the design of public policies. Their exclusion leads to ineffective policies

Author Response

Thank you for you review, it was very helpful and guided us to improve the scientific quality of the paper. I hope we have addressed the issues you presented satisfactorily.

Improvements on research design and methods and the sampling issue:

We have expanded the material and methods sections to describe in much more detail the steps of conducting the thematic content analysis of the gathered materials and elaborated on the role of the experts as facilitators of knowledge exchange throughout this process.  The main objective as presented in the research questions (row 196->) is to illustrate, qualitatively, the different ways that European citizens discuss environmental concerns. The issue of research design from the perspective of representativeness and recognizing that it is impossible to gather a representative sample of European citizens to discuss these types of questions together is now highlighted throughout the text, with the introduction discussing the problem of representativeness, the material and methods coming back to this from the perspective of reaching different publics (row 208->) and finally in the discussion to elaborate on the possible consequences of the things discussed in the workshops because of a certain type of group of citizens participating (row 607->).

Conclusions:

We have edited the conclusions so that they summarise the main findings of our work and highlight based on the results and discussions the responses to our research questions. We wanted to keep the conclusions brief and concise.

We have also done other changes based on the other reviewers comments, particularly expanding the discussion to analyse the results further. All changes are done with the track-changes function, except minor language-corrections.

Happy holidays!

Reviewer 2 Report

Dear Authors,

I revised the manuscript "An empirical analysis of citizen engagement for the EU post-2020 biodiversity strategy" submitted to the Sustainability Journal. The manuscript is very interesting, addressing the biodiversity strategy after 2020 in the EU, but now isn’t a scientific work. In my opinion it is a case study containing "feature paper" elements, mainly presents the methodological approach and citizen expectations and reflections.

I have some concerns which need to be addressed before considering for final publication in present form. The manuscript should be improved in sections Materials and Methods, Results and Discussion in areas: research design, presented results. Full experimental and methodical details must be provided so that the results can be reproduced.

However, if the authors decide to change the manuscript type to another one, e.g. "Perspective" or "Concept Paper" or to add "Article - Feature Paper", then the manuscript with my comment is suitable for publication.

Specific remarks:

Line 127. When you first use abbreviation "EKLIPSE", explain it.

Author Response

Dear reviewer, 

Thank you for your valuable comments. We have now revised the paper and done major revision to the different sections. Please also see the cover letter attached.

Research design and material and methods, results:

We elaborated on how the expert working group who designed and produced the whole research process came and worked together. Most importantly we describe step by step how the qualitative thematic content analysis of the workshop materials was conducted and why workshops are an appropriate research methodology for this type of research and why/how the expert working group members worked as facilitators of knowledge exchange throughout the process. We feel that with these additions it is clearer how we obtained the results. The results of section 3 are a pure representation of the outcomes of the workshops based on the content analysis conducted.

Discussion:
We have elaborated the discussion to have more detailed analysis and reflections of the results based on the scientific literature relevant to the field of citizen participation in (environmental) science and policy processes. Previously the paper was rather narrow in the discussions and we feel that by these elaborations, and those in material and methods, the scientific quality of the paper has increased making it a scientific work and therefore maintain to keep this as an article format for this special issue.

The specific remark of explaining EKLIPSE has also been addressed.

Happy holidays!

Round 2

Reviewer 1 Report

The answer provided by the authors in their cover letter to my main objection (“In section 4.2 some limitations on having more women andurban residents are acknowledged and the challenges of encouraging participation from the perspectiveof costs of participating in these types of knowledge exchange activities is presented”) to the publication of the article seems completely insufficient to me. In fact in paragraph 4.4 (lines 607-618) the same authors highlight the limits of this approach. (“Reflecting on this principle, it can be fairly argued that our workshops did not capture citizens from diverse domains some of which may be more affected by the biodiversity policy, including  landowners, businesses, rural habitants etc. and who may also hold relevant knowledge to discover and provide acceptable solutions to the environmental issues”).

Author Response

Dear reviewer,

Thank you for your response. From the start of designing our research we had decided that we would use workshops as our data collection method, since the objective was to encourage dialogue and see how European citizens discuss environmental issues. We knew that open workshops as a method would not necessarily gather a representative or balanced sample of genders, professions, socio-economic demographics, urban/rural residents etc. and this was never the objective in the research design and this and the possible limitations of the method and workshops are now clearly stated in various parts of the paper largely thanks your original revision request. Our study is qualitative study and acknowledging this we do not make any claims in the paper that the views from the workshop wholly represent views of all Europeans. If this was a quantitative study where we would’ve aimed to get a representative sample and analyse also differences between different social groups, your critique would be valid. Research and studies will always have to make decisions regarding scope and the limitations they may produce but as long as we are transparent about these, we have conducted an ethical and appropriate qualitative examination. One of the conclusions and findings of the paper is that we need to also put future efforts in engaging people from different social backgrounds and those ignorant to environmental issues. As these notions are in the paper, we do not see that there is a need for further revisions, nor would it be possible to influence the research design at this stage to satisfy the missing perspectives by e.g. hosting more workshops since this would more likely skew the results rather than help the paper.

Hereby we leave the decision of publication to the editors of the special issue.

Reviewer 2 Report

From my point of view, the mistakes were corrected and the resubmitted manuscript was strongly improved.

Author Response

Thank you